# Prior-aware and Context-guided Group Sampling for Active Probabilistic Subsampling

**Beomgu Kang, Hyunseok Seo**[*]
Department of Artificial Intelligence, Korea University, Seoul, Republic of Korea
`bgkang1@korea.ac.kr, seoh@korea.ac.kr`

## Abstract

Subsampling significantly reduces the number of measurements, thereby streamlining data processing and transfer overhead, and shortening acquisition time across diverse real-world applications. The recently introduced Active Deep Probabilistic Subsampling (A-DPS) approach jointly optimizes both the subsampling pattern and the downstream task model, enabling instance- and subject-specific sampling trajectories and effective adaptation to new data at inference time. However, this approach does not fully leverage valuable dataset priors and relies on top-1 sampling, which can impede the optimization process. Herein, we enhance A-DPS by integrating a deterministic (fixed) prior-informed sampling pattern derived from the training dataset, along with group-based sampling via top-k sampling, to achieve more robust optimization—a method we call Prior-aware and context-guided Group-based Active DPS (PGA-DPS). We also provide a theoretical analysis supporting improved optimization via group sampling, and validate this with empirical results. We evaluated PGA-DPS on three tasks: classification, image reconstruction, and segmentation, using the MNIST, CIFAR-10, fastMRI knee, and hyperspectral AeroRIT datasets, respectively. In every case, PGA-DPS outperformed A-DPS, DPS, and all other sampling methods. Our code is available at https://github.com/B9Kang/PGADPS.

## 1 Introduction

Modern technologies generate massive datasets that can require lengthy acquisition time and hinder real-time onboard processing. Many real-world applications, including Magnetic Resonance Imaging (MRI) (Ye, 2019), computed tomography (CT) imaging (Chen et al., 2008), ultrasound imaging (Huijben et al., 2020b), digital micromirror device (Baraniuk, 2007), seismic surveying (Herrmann et al., 2012), and hyperspectral imaging (Sun & Du, 2019), highlight the importance of reducing imaging data volume. Therefore, strategic sampling not only reduces data volume and preserves essential information for efficient transfer and processing but also accelerates acquisition, a critical factor in medical imaging.

Compressed sensing (CS) was developed as a subsampling strategy to overcome the Nyquist-Shannon limits on the sampling rates required for perfect signal reconstruction (Donoho, 2006; Eldar & Kutyniok, 2012). CS has been widely adopted across various applications and has demonstrated significant impact (Lustig et al., 2007; Baraniuk & Steeghs, 2007; Yu & Wang, 2009; Martín et al., 2014; Lorintiu et al., 2015; Han et al., 2016). Although CS exploits the inherent signal structures, such as sparsity, it does not take into account the information relevant to the downstream task during the sampling process. Bridging the gap between modality- and task-specific knowledge and the sampling process has proven challenging.

Recently, subsampling techniques customized for specific data distributions and downstream tasks have been proposed with advances in deep learning, offering learned yet fixed sampling patterns (Huijben et al., 2020a; Weiss et al., 2020; Shen et al., 2020; Sherry et al., 2020; Zhang et al., 2020; Aggarwal & Jacob, 2020; Mou et al., 2021; Yang et al., 2025). These approaches optimize a sampling pattern based on the average data distribution in the training set, which may not provide optimal results for individual instances. To address this limitation, active sampling was introduced,

---

[*]Corresponding author

where new points are adaptively selected based on previously acquired samples, iterating until the required target number of samples is obtained (Zhang et al., 2019; Jin et al., 2019; Bakker et al., 2020; Pineda et al., 2020; Van Gorp et al., 2021; Tian et al., 2025). Active sampling produces a sampling trajectory that adapts to each test instance and incorporates newly acquired data dynamically. This is particularly valuable in medical imaging, where each patient's unique physiological condition demands individualized acquisition trajectories.

Although active subsampling provides instance-specific sampling patterns and enables effective adaptation to new data, boosting downstream task performance, its sampling strategy still has room for improvement. Through iterative top-1 sampling guided by previously selected samples, active sampling leverages inter-sample relationships to optimize the sampling model and minimizes redundant information. However, it fails to fully exploit the rich prior knowledge embedded in the training dataset. Moreover, relying on top-1 sampling leads to sub-optimal optimization (Huijben et al., 2020a).

In this work, we enhance Active Deep Probabilistic Subsampling (A-DPS) architecture by incorporating deterministic (fixed) subsampling based on prior knowledge of the training data with group sampling to achieve more robust optimization (Fig. 1). We call this approach Prior-aware and context-guided Group-based Active Deep Probabilistic Subsampling (PGA-DPS). We demonstrate that PGA-DPS exploits training-data priors via deterministic sampling and reinforces this with instance-adaptive group sampling, resulting in a smoother loss landscape and more stable optimization. We evaluated PGA-DPS on the MNIST dataset (LeCun et al., 1998), the CIFAR-10 dataset (Krizhevsky et al., 2009), the fastMRI knee dataset (Zbontar et al., 2018), and the hyperspectral AeroRIT dataset (Rangnekar et al., 2020). Across classification, image reconstruction, and segmentation tasks, PGA-DPS outperforms all other state-of-the-art sampling methods.

## 2 RELATED WORK

Recent works have proposed learning-based subsampling techniques tailored to data types and downstream tasks. In the field of MRI, Learning-based Optimization of the Under-sampling PattErn (LOUPE) was developed to simultaneously optimize the subsampling pattern and reconstruct the image, addressing two core challenges of compressed sensing (Bahadir et al., 2020). LOUPE learns the sampling mask by relaxing the non-differentiable threshold operation, which is similar to the Gumbel-softmax trick used in Deep Probabilistic Subsampling (DPS) (Huijben et al., 2020a). The target sampling ratio is enforced by an L1 sparsity penalty on sampling mask within the loss function. Additionally, a stochastic greedy algorithm has been applied in MRI to obtain the sampling pattern that minimizes the reconstruction loss (Sanchez et al., 2020). However, greedy algorithm-based methods require pre-trained reconstruction capable of handling any subsampling pattern to thoroughly evaluate the loss.

In hyperspectral imaging (HSI), band selection techniques are widely used to identify a small informative subset of spectral bands and reduce spectral redundancy. Self-Representation Learning with Sparse 1D-Operational Autoencoder (SRL-SOA) employs a sparse autoencoder model to learn a sparse representation using a Taylor-series expansion of non-linear transformation (Ahishali et al., 2022). However, this does not incorporate the downstream task model for band selection. Greedy Spectral Selection (GSS) adopts a greedy algorithm: it first performs filter-based interband redundancy analysis, then trains a Convolutional Neural Network (CNN) to assess classification performance of the chosen bands (Morales et al., 2021). On the other hand, CNN based on Bandwise-independent convolution and Hard thresholding (BHCNN) jointly optimizes band selection and the classification by applying a hard threshold mask via a straight-through estimator (Feng et al., 2020). Although this allows joint optimization, gradients for non-selected bands are ignored, which can hinder the discovery of other informative bands during training. Furthermore, the aforementioned task-aware band selection methods are designed for classification, which may not provide optimal spectral bands for segmentation tasks.

An active sampling strategy for MRI k-space acquisition uses an adversarial neural network to distinguish between sampled and unsampled lines in the Fourier space (Zhang et al., 2019). The adversarial network identifies the k-space line that appears most realistically fake and selects it for acquisition; this process repeats until the desired number of lines is obtained. However, this approach processes k-space data directly, which limits its applicability to MR image undersampling.

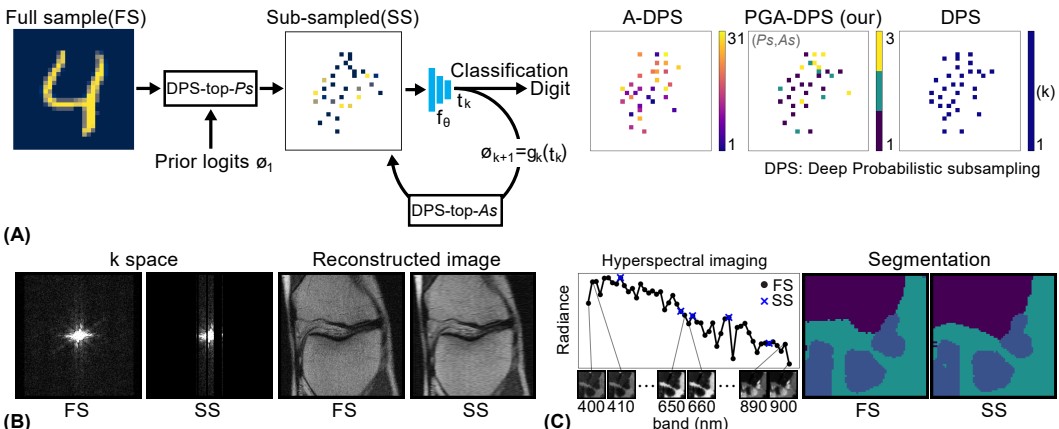

Figure 1: **(A)** A schematic overview of the proposed Prior-aware and Group-based Active DPS (PGA-DPS) applied to classification task on the MNIST dataset. PGA-DPS uses learned prior logits ($\phi_1$), for a fixed sampling mask and then acquires new samples in grouped, active iterations. Here, $Ps$ and $As$ stand for portions of prior and active sampling respectively. For example, DPS picks 31 samples in one step, A-DPS over 31 iterations, and PGA-DPS in just 3 iterations. **(B)** An example of an MRI reconstruction task. **(C)** An example of an HSI segmentation task.

Reinforcement learning (RL) has also been explored for active acquisition, applying greedy and modified $\epsilon$-greedy policies to actively acquire optimal sampling masks (Pineda et al., 2020; Bakker et al., 2020). However, RL-based techniques require a pretrained reconstruction network that can handle a variety of sampling patterns. To address this issue, Active DPS (A-DPS) jointly optimizes both the sampling mask and the associated reconstruction network (Van Gorp et al., 2021). Building on A-DPS, the proposed Prior-aware and Group-based Active DPS (PGA-DPS) improves optimization robustness and fully leverages prior knowledge embedded in the training data.

## 3 METHOD

### 3.1 TASK-ADAPTIVE SUBSAMPLING FRAMEWORK

We aim to find an optimal subsampling strategy $A \subseteq \{0, 1\}^N$ on a given input signal $x \in \mathbb{R}^N$ for a specific task $t$, such that the performance of the task is maintained using fewer input samples.

$$\min_{A,\theta} \left\| t - \hat{t}(\theta, A) \right\| \quad \text{subject to} \quad \sum_{i=1}^{N} A_i = M, \quad \text{where } A_i \in A \tag{1}$$

$$\hat{t}(\theta, A) = f_{\theta, A}(Ax) \tag{2}$$

where $f$ is a task model that predicts the task $t$ from the input signal $x$, $\theta$ is parameters of the model, and $M$ is the target number of subsamples. The proportion of selected samples can be represented by the sampling ratio $r = 100 \times M/N\%$. In general, neural networks are widely adopted for task models, such as classification, segmentation, reconstruction, detection, and more; however, backpropagation from the loss function is hindered by the non-differentiable nature of sampling. This issue was alleviated using the Gumbel-Softmax reparameterization trick introduced in Deep Probabilistic Subsampling (DPS).

### 3.2 DPS: DEEP PROBABILISTIC SUBSAMPLING

DPS proposed an end-to-end deep learning framework that jointly optimizes both the subsampling strategy and the downstream task model (Huijben et al., 2020b). During training, DPS learns unnormalized logits $\phi$, which are used to generate a subsampling mask probability $P(A|\phi)$ from a categorical distribution with $N$ classes: $A \sim \text{Categorical}\left(\frac{\exp(\phi_i)}{\sum_j^N \exp(\phi_j)}, i \in \{1, \ldots, N\}\right)$. The

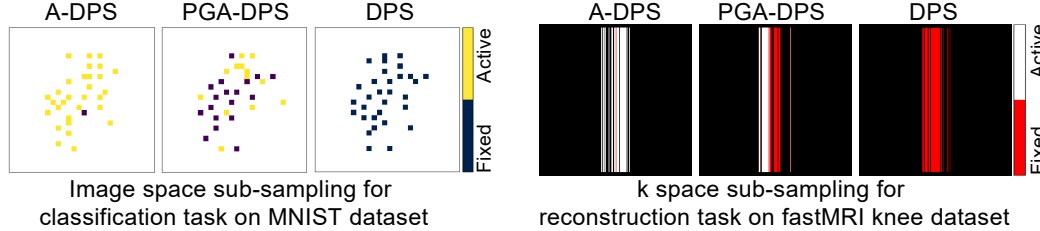

Figure 2: An illustration of active sampling strategy for A-DPS and PGA-DPS, whereas a fixed (deterministic) masking pattern is used for DPS.

Gumbel-max trick enables efficient sampling from a categorical distribution (Gumbel, 1954; Maddison et al., 2014):

$$A_i = \begin{cases} 1 & \text{if } i \in \arg\max G_{\phi_i} \\ 0 & \text{otherwise} \end{cases} \tag{3}$$

$$G_{\phi_i} = G_i + \phi_i \sim \text{Gumbel}(\phi_i), \; i \in \{1, \dots, N\} \tag{4}$$

where $\phi \in \mathbb{R}^N$ is an unnormalized logit vector composed of $\phi_i$ and $G$ is a vector of i.i.d noise samples, $G_i$, drawn from a Gumbel distribution $\sim$ Gumbel$(0, 1)$. However, since the arg max function is non-differentiable and thus unsuitable for backpropagation, DPS employs the Gumbel-Softmax trick by relaxing the max operation into a differentiable softmax function as follows (Jang et al., 2017; Maddison et al., 2017):

$$\nabla_\phi A := \nabla_\phi softmax_\tau (G_\phi) \tag{5}$$

where $\tau$ is a temperature parameter that controls the degree of relaxation, enabling gradients to be distributed across multi-dimensional logits when $\tau > 0$ during training. In addition, as the temperature decreases $\tau \to 0$, $softmax_\tau$ can anneal into a categorical distribution. Although tuning the relaxation with $\tau$ is crucial for managing the bias-variance trade-off of the gradient estimator (Tucker et al., 2017), the DPS architecture has been shown to perform effectively with a fixed temperature during training ($\tau = 2$).

In addition, the Gumbel-max trick can be extended to the Gumbel top-$k$ trick to efficiently select the $k$ most probable samples, rather than just the single highest-probability one (Kool et al., 2019; Plötz & Roth, 2018).

$$A = \arg \underset{k}{\text{top}} \, (\phi + G) \tag{6}$$

DPS introduces two variants for selecting $k$ samples: DPS-top-1, which sequentially samples from $k$ independent categorical distributions—each with its own trainable logits—and DPS-top-$k$, which selects all $k$ samples simultaneously using a single shared logit for greater parameter efficiency. Although DPS-top-1 is more expressive, the DPS-top-$k$ approach showed improved performance.

Similarly, Active Deep Probabilistic Subsampling (A-DPS) introduces an active sampling that uses contextual information from previously selected samples to guide subsequent selection, allowing adaptation to new data (Van Gorp et al., 2021). In A-DPS, samples are chosen iteratively by applying DPS-top-1 at every step.

$$A^j = \arg \underset{1}{\text{top}} \left\{ w^{j-1} + \phi^j + G^j \right\}, \quad j = 1, 2, \dots, K \tag{7}$$

$$\phi^j = g^j(t^j), \quad t^j = f_\theta(A^{j-1}x) \tag{8}$$

where $j$ denotes the iteration; $w^j \subseteq \{-\infty, 0\}^N$ is a cumulative mask that assigns minus infinity to previously selected elements, ensuring they are excluded after re-normalization; $\phi^j \in \mathbb{R}^N$ and $G^j \in \mathbb{R}^N$ are $\phi$ and $G$ for $j^{th}$ sample, respectively; $g^j$ is a sampling network in $j^{th}$ iteration that encodes the current task context based on the samples selected so far; and $f_\theta$ is a differentiable model that generates the current task prediction. The context is built using an *analysis-by-synthesis* approach, where the task (synthesis) module guides the sampling (analysis). In A-DPS, losses are accumulated over all iterations, and the network is updated in a semi-greedy manner.

### 3.3 PGA-DPS: PRIOR-AWARE AND GROUP-BASED ACTIVE DPS

PGA-DPS initiates with a deterministic subsampling pattern and then iteratively acquires additional samples through active sampling (Fig. 2). The deterministic pattern drawn from the training data's prior is followed by active sampling that exploits each new input's contextual information, thereby combining global data prior with input specific context.

Moreover, PGA-DPS uses DPS-top-$k$ to select samples in groups for active sampling, rather than DPS-top-1 in A-DPS, enabling it to achieve a smaller effective Lipschitz constant. We analyze group sampling enabled by DPS-top-$k$ in terms of the loss, which captures the Lipschitzness of the loss and reflects the smoothness of the optimization landscape. The proof is available in Appendix A.

**Theorem 1.** *(The effect of group sampling on the Lipschitzness of the loss). Let $f_1, f_2, \ldots, f_k$ be task models, where each function $f_j$ has a corresponding Lipschitz constants $L_j$ for $j = 1, 2, \ldots, k$. In DPS-top-1, k task functions are required for each input sample $x_k$, whereas DPS-top-k defines a single task function $f_k$ over group of k samples. The Lipschitzness of the loss function for each DPS-top-1 and DPS-top-k is as follows:*

$$\|Loss_{DPS\text{-}top\text{-}1}(x)\|^2 = \|f(x^*) - f(x)\|^2 \leq \prod_{r=1}^{k} L_r \|x_1^* - x_1\|^2, \quad f = f_k(f_{k-1}(\ldots(f_1(x_1))\ldots))$$

$$\|Loss_{DPS\text{-}top\text{-}k}(x)\|^2 = \|f_k(x^*) - f_k(x)\|^2 = \|f_k(x^*) - f_k(x_1, x_2, \ldots, x_k)\|^2 \leq L_k \|x^* - x\|^2$$

*Here, $x^* \in \mathbb{R}^N$ is the fully sampled input signal (unsampled signal = 0), $x \in \mathbb{R}^N$ is the sampled input signal, and $x_r$ is the $r^{th}$ sample. If the Lipschitz constant of task model at each iteration $L_j \geq 1$,*

$$L_k \leq \prod_{r=1}^{k} L_r \implies \sup_x \|Loss_{DPS\text{-}top\text{-}k}(x)\|^2 \leq \sup_x \|Loss_{DPS\text{-}top\text{-}1}(x)\|^2$$

Neural networks typically have Lipschitz constants much greater than one, except in the trivial near-identity case (Malherbe & Vayatis, 2017; Bartlett et al., 2018; Latorre et al., 2020; Shi et al., 2022). Because our task of classification, image reconstruction (from k-space to image), and segmentation, are far from identity mappings, DPS-top-$k$ exhibits a smaller effective Lipschitz constant than DPS-top-1, whose effective constant is the product of multiple Lipschitz constants. Moreover, the same analysis can apply to the Lipschitz constants of gradient loss $\|\nabla Loss\|$. Consequently, PGA-DPS creates smoother latent space, resulting in more stable and efficient optimization (Virmaux & Scaman, 2018; Berkenkamp et al., 2017; Liu et al., 2022; Gouk et al., 2021; Santurkar et al., 2018). This theoretical analysis aligns with previous experimental results that Top-$k$ sampling outperformed Top-1 sampling in DPS (Huijben et al., 2020a).

## 4 EXPERIMENTS

### 4.1 MNIST CLASSIFICATION

#### 4.1.1 EXPERIMENT SETUP

We evaluated the classification performance under pixel subsampling using the MNIST dataset (Le-Cun et al., 1998). The 70,000 grayscale images of $28 \times 28$ pixels, representing handwritten digits from 0 and 9, were split into train, validation, and test sets with a ratio of 5:1:1, respectively. We compare the proposed PGA-DPS method with DPS and A-DPS across various sampling ratios, ranging from 1% to 8% with the increment of 1%. Furthermore, we also report the reference performance obtained using all samples (100%).

#### 4.1.2 TASK MODEL

A multi layer perceptron (MLP) with 5 layers is used for the classification network $f_\theta(\cdot)$. The MNIST images are flattened and masked according to the subsampling pattern before being fed into the network. The sampling network $g^k(\cdot)$ of an LSTM (Long Short-Term Memory) followed by a two-layer MLP, which encodes the current task context and generates the corresponding sampling

mask. Both the classification network and sampling network are simultaneously trained using a categorical cross-entropy loss. In addition, the trainable logits of size 784 ($\phi_1$) are included in the network for DPS and PGA-DPS to provide a deterministic sampling pattern. Detailed model architecture and training settings are provided in the Appendix B.1.

In PGA-DPS, the proportions of prior (deterministic) sampling and active sampling are fixed to 60 and 20 %. Here, the $Ps$ and $As$ denote the proportions of prior and active sampling, respectively. ($Ps$, $As$)=(60, 20) implies that 60% of the target samples are selected by prior sampling, and the remaining samples (40%) are acquired over two active group sampling iterations of 20% each.

### 4.1.3 RESULTS

The classification accuracy on test set is shown in Table 1. PGA-DPS outperforms both DPS and A-DPS across all sampling ratios, with the largest gain at the low sampling ratios. The active sampling method leverages previously selected samples for adapting its sampling pattern, which is particularly effective under extreme subsampling conditions. A-DPS showed strong performance when sampling ratio ($r$) is below 4%, whereas DPS works better when more samples are available. This likely arises because A-DPS trains a separate classifier for each sampled pixel, inflating its Lipschitz constant (as discussed in Theorem 1).

Table 1: The classification accuracy on the MNIST test set (10,000 samples) for various subsampling ratios ($r$). Each sampling strategy was repeated across six independent runs.

| Sampling Model | Sampling ratio: r (%) | | | | | | | | |
|---|---|---|---|---|---|---|---|---|---|
| | 1 | 2 | 3 | 4 | 5 | 6 | 7 | 8 | 100 |
| DPS | 63.5 | 83.6 | 91.5 | 95.2 | 96.4 | 97.1 | 97.3 | 97.6 | 98.2 |
| A-DPS | 64.6 | 85.2 | 92.2 | 95.3 | 96.1 | 96.6 | 97.1 | 97.2 | - |
| PGA-DPS | **68.8** | **86.9** | **93.9** | **95.8** | **96.7** | **97.2** | **97.5** | **97.7** | - |

## 4.2 CIFAR-10 CLASSIFICATION

### 4.2.1 EXPERIMENT SETUP

We evaluated pixel-level subsampling for classification using the CIFAR-10 dataset (Krizhevsky et al., 2009), which contains 60,000 color images of size 32 x 32 across 10 classes. Of these, 50,000 images were used for training and validation, and 10,000 for testing. The training split was further divided into training and validation set at a 9:1 ratio. We compared the proposed PGA-DPS, DPS and A-DPS methods across sampling ratios ranging from 2% to 20% in increments of 2%. The reference performance using all samples (100 %) is also reported.

### 4.2.2 TASK MODEL

For the classification model $f_\theta(\cdot)$, we employed a four-layer convolutional neural network (CNN) followed by three MLP layers. The CIFAR-10 images are first masked according to the subsampling pattern and then fed into the network. We employed the same sampling network architecture and the same cross-entropy loss function as in the MNIST classification experiments. The detailed model architecture and training settings are provided in the Appendix B.2. In PGA-DPS, the proportions of prior sampling and active sampling are fixed to 10 % and 20 %, respectively.

Table 2: The classification accuracy on the CIFAR-10 test set (10,000 samples) for various subsampling ratios ($r$). Each sampling strategy was repeated across six independent runs.

| Sampling Model | Sampling ratio (%) | | | | | | | | | | |
|---|---|---|---|---|---|---|---|---|---|---|---|
| | 2 | 4 | 6 | 8 | 10 | 12 | 14 | 16 | 18 | 20 | 100 |
| DPS | 38.6 | 44.5 | 43.4 | 44.1 | 46.1 | 46.5 | 49.2 | 50.5 | 52.1 | 54.3 | 96.3 |
| A-DPS | 52.9 | 61.0 | 65.8 | 67.4 | 70.4 | 69.6 | 70.7 | 69.5 | 70.0 | 68.3 | - |
| PGA-DPS | **54.3** | **63.2** | **68.6** | **70.8** | **74.7** | **78.1** | **78.6** | **79.8** | **81.3** | **82.7** | - |

### 4.2.3 RESULTS

Table 2 shows that the proposed PGA-DPS outperforms both DPS and A-DPS across all sampling ratios. Because CIFAR-10 classification is substantially more challenging than MNIST, the performance gains are larger across the entire range. Unlike the MNIST classification task, A-DPS exhibits consistently strong performance relative to DPS for all ratios. This behavior is likely attributable to the lower Lipschitz constant of CNNs compared to MLPs used in the MNIST experiments (Shi et al., 2022), which makes the optimization landscape easier to navigate. However, beyond a certain number of samples ($r = 14 \%$), the classification accuracy of A-DPS decreased, likely due to an inflated Lipschitz constant, similar to what was observed in the MNIST classification task.

## 4.3 MRI RECONSTRUCTION

### 4.3.1 EXPERIMENT SETUP

To evaluate the performance of k-space subsampling, we tested on the single-coil knee RAW acquisitions from the fastMRI dataset (Zbontar et al., 2018). A total of 13,000 images, excluding the outer slices, were split into an 8:2:3 ratio for train, validation, and test, respectively. All slices were cropped to central $208 \times 208$ pixels and normalized to [0, 1]. Vertical Cartesian binary masks ($\mathbf{M}$) were used for all sampling models, where one column line corresponds to acquiring one phase-encoding line. We simulated the partially acquired k-space by applying the Cartesian binary mask to a fully sampled k-space:

$$\mathbf{Y} = \left| \mathcal{F}^{-1} \mathbf{M} \mathcal{F} \mathbf{X} \right| \tag{9}$$

where $\mathcal{F}, \mathcal{F}^{-1}$ are the forward and inverse Fourier transforms, respectively; $\mathbf{Y} \in \mathbb{R}^{N \times N}$ is the image from subsampled k-space; $\mathbf{X} \in \mathbb{R}^{N \times N}$ is the ground truth image from fully sampled k-space; and $|\cdot|$ denotes the magnitude operation. Following the previous study's setup (Van Gorp et al., 2021), we take the magnitude of complex-valued Y to simplify the image reconstruction.

### 4.3.2 TASK MODEL

For the reconstruction task $f_\theta(\cdot)$, a deep unfolded proximal gradient method is employed to reconstruct the original image $\hat{\mathbf{X}}$ from the partially acquired measurement $\mathbf{Y}$ (Mardani et al., 2018), by unrolling iterations of the proximal gradient algorithm into a feed-forward neural network:

$$\hat{\mathbf{X}}^{k+1} = \mathcal{P}_\psi \left( \hat{\mathbf{X}}^k - \alpha \nabla \left| \mathbf{Y} - \mathcal{F}^{-1} \mathbf{M} \mathcal{F} \mathbf{X} \right| \right) \tag{10}$$

where $\mathcal{P}_\psi$ denotes the proximal neural network incorporating the image prior regularizer ($\psi$), and $\alpha$ is the step size. The sampling network $g^k(\cdot)$ consists of a 3-layer CNN, followed by global average pooling, an LSTM, and a single MLP, in sequence. In this way, the current reconstruction result is encoded into the LSTM to generate logits ($\phi$), which are used to select the subsequent (active) sample. The reconstruction model and training details are provided in the Appendix B.3.

We compare PGA-DPS to other sampling methods, namely, low-pass, variable density sampling (VDS) (Lustig et al., 2007), greedy mask selection (Sanchez et al., 2020), LOUPE (Bahadir et al., 2020), DPS, and A-DPS. In the loss-pass method, lines nearest to the DC (k-space origin) components are selected; in VDS, sampling density decays with distance from the k-space origin, concentrating the samples centrally and tapering off toward the periphery. In the greedy mask strategy, the mask pattern is optimized using a stochastic greedy algorithm (Sanchez et al., 2020), implemented via the NESTA solver (Becker et al., 2011). Afterwards, the corresponding proximal gradient network was trained using the fixed optimal mask obtained from the greedy algorithm (Greedy Prox.).

### 4.3.3 RESULTS

To assess the influence of the sampling hyperparameters (*Ps, As*), we conducted an ablation study using their various combinations. Table 3 demonstrates the MR reconstruction performance on the hold-out test set, evaluated using three metrics: normalized mean square error (NMSE), peak signal-to-noise ratio (PSNR), and structural similarity index (SSIM). Reconstruction performance shows an overall increasing trend with higher *As* (i.e., larger k in top-k sampling), across various levels of prior sampling (*Ps*), even without the use of prior-aware deterministic sampling, which is equivalent

to A-DPS with group sampling. This trend is consistent with the theoretical explanation proposed in Theorem 1. However, the improvement plateaus beyond an *As* of 15-20% , or even declines slightly, presumably due to the reduced rate of active sampling. While PGA-DPS shows slight performance variation across (*Ps, As*) configurations, it outperforms A-DPS in most cases.

We found the (30%, 30%) configuration of (*Ps, As*) to be the optimal and fixed it for further MRI reconstruction analysis. Furthermore, this optimal configuration generalizes well across different sampling ratios of k-space lines M = 15, 20, 30, and 35 (Table 8 in the Appendix). The performance gain is particularly notable in the low-measurement regime, where the number of k-space lines is small, further demonstrating the robustness of our method under limited data conditions.

Table 3: Average results over 10 runs on the hold-out test set of size $208 \times 208$ pixels for an acceleration factor of 8 ($M = 26$, $r = 12.5\%$), using PGA-DPS across various $Ps$ and $As$.

|              |      | $As = 5\%$ | 10%    | 15%    | 20%    | 30%    | 40%    |
|--------------|------|--------|--------|--------|--------|--------|--------|
|              | NMSE | 0.0398 | 0.0393 | 0.0389 | 0.0373 | 0.0376 | 0.0387 |
| $Ps = 0\%$   | PSNR | 25.4   | 25.4   | 25.5   | 25.7   | 25.6   | 25.5   |
|              | SSIM | 0.576  | 0.581  | 0.584  | 0.598  | 0.591  | 0.585  |
|              | NMSE | 0.0388 | 0.0393 | 0.0378 | 0.0370 | **0.0359** | 0.0359 |
| $Ps = 30\%$  | PSNR | 25.5   | 25.4   | 25.6   | 25.8   | **25.9** | 25.9   |
|              | SSIM | 0.588  | 0.585  | 0.597  | 0.618  | **0.621** | 0.619  |
|              | NMSE | 0.0387 | 0.0377 | 0.0372 | 0.0364 | 0.0366 | 0.0367 |
| $Ps = 50\%$  | PSNR | 25.6   | 25.6   | 25.7   | 25.9   | 25.8   | 25.8   |
|              | SSIM | 0.592  | 0.598  | 0.603  | 0.615  | 0.610  | 0.611  |
|              | NMSE | 0.0370 | 0.0377 | 0.0368 | 0.0375 | 0.0371 | —      |
| $Ps = 70\%$  | PSNR | 25.7   | 25.7   | 25.8   | 25.7   | 25.7   | —      |
|              | SSIM | 0.596  | 0.600  | 0.606  | 0.601  | 0.600  | —      |

As shown in Table 4, the proposed PGA-DPS significantly outperforms all other sampling strategies across all evaluation metrics. The example reconstruction from masks generated by PGA-DPS, A-DPS, DPS, and other sampling methods are shown in the Appendix (Fig. 3).

Since DPS optimizes its mask to perform well on average, it heavily concentrates on DC (low-frequency) lines, that capture overall structure of the input data (Fig. 2). In contrast, PGA-DPS and A-DPS select more samples farther from the DC region, capturing richer structural details. Note that A-DPS always begins at the center line, whereas PGA-DPS starts with a mix of central and peripheral lines, achieving a more balanced representation of frequency components.

### 4.4 MRI RECONSTRUCTION WITH ACTIVE BASELINES

#### 4.4.1 EXPERIMENT SETUP

To further validate the effectiveness of our method, we conducted an additional comparison against reinforcement learning (RL)-based sampling strategies: Evaluator Policy (Zhang et al., 2019) and Data-Specific Double Deep Q-Networks (DS-DDQN) (Pineda et al., 2020). To ensure compatibility with the publicly available DS-DDQN checkpoints, we applied the same pre-processing procedures used in their original work. In addition, the trained checkpoints were used for DPS (DPS_16lines_0seed_Pineda) and A-DPS (ADPS_16lines_0seed_Pineda). We adopt the train-validation-test split protocol from (Van Gorp et al., 2021), yielding 34,742 training, 1,785 validation, and 1,851 testing images. Each input corresponds to a k-space volume of size 368×640. For evaluation, reconstruction scores are computed over the central 320×320 region, excluding peripheral background areas. PGA-DPS was trained for 5 epochs with a batch size of 1, following the training scheme provided in A-DPS (Van Gorp et al., 2021). The Ps and As ratios were both set to 30%.

#### 4.4.2 RESULTS

Table 5 presents the MR reconstruction results on the hold-out test set with an acceleration factor of 8. We adopt the Scenario-30L setting, in which 30 Auto-Calibration Signal (ACS) lines are fixed, and the remaining 16 target samples are selected. PGA-DPS outperforms all other sampling methods

Table 4: Average results over 10 runs on the hold-out test set of size 208×208 pixels for an acceleration factor of 8 ($M = 26, r = 12.5\%$) compared to other sampling methods.

| | | | Sampling Model | | | | |
|---|---|---|---|---|---|---|---|
| | Low pass | VDS | Greedy Prox. | LOUPE | DPS | A-DPS | PGA-DPS |
| NMSE | 0.0462 | 0.0471 | 0.0425 | 0.0465 | 0.0408 | 0.0398 | **0.0359** |
| PSNR | 24.5 | 24.5 | 25.0 | 25.1 | 25.3 | 25.4 | **25.9** |
| SSIM | 0.511 | 0.533 | 0.536 | 0.574 | 0.571 | 0.576 | **0.621** |

under this setting. Notably, the overall PSNR and SSIM achieved in this experiment is substantially higher than that reported in our previous MRI study using a smaller matrix size (208x208), reflecting the benefits of using higher-resolution k-space data with abundant ACS lines.

Table 5: Results on the hold-out test set of size $368 \times 640$ pixels for an acceleration factor of 8 ($M = 46, r = 12.5\%$, ACS = 30) compared to RL approaches.

| Sampling Method | NMSE | PSNR | SSIM |
|---|---|---|---|
| Evaluator policy (Zhang et al., 2019) | 0.0398 | 28.8 | 0.610 |
| DS-DDQN (Pineda et al., 2020) | 0.0370 | 29.2 | 0.623 |
| DPS (Huijben et al., 2020a) | 0.0360 | 30.1 | 0.650 |
| A-DPS (Van Gorp et al., 2021) | 0.0342 | 30.2 | 0.654 |
| PGA-DPS (Proposed) | **0.0331** | **30.5** | **0.668** |

## 4.5 HYPERSPECTRAL IMAGE (HSI) SEGMENTATION

### 4.5.1 EXPERIMENT SETUP

To analyze the generalizability of PGA-DPS, we evaluated its performance on hyperspectral band selection for segmentation. We used the AeroRIT dataset (Rangnekar et al., 2020), which consists of aerial images with hyperspectral bands from 400 to 900 nm in 10 nm increments, resulting in a total of 51 spectral bands. Following the original study's split, $1920 \times 3968$ image was divided into the left $1920 \times 1728$ for training, the next $1920 \times 512$ for validation, and the final $1920 \times 1728$ for testing. We extracted $64 \times 64$ patches with 50% overlap from the training set and non-overlapping patches for validation and test sets. The segmentation labels include five classes : (1) roads, (2) buildings, (3) vegetation, (4) cars, and (5) water.

### 4.5.2 TASK MODEL

For the segmentation network $f_\theta(\cdot)$, we use a residual U-Net consisting of 6 ResNet blocks following the pix2pix design (Isola et al., 2017; Zhu et al., 2017): this architecture is referred to as Res-U-net in the AeroRIT study (Rangnekar et al., 2020). The sampling network $g^k(\cdot)$ consists of a 4-layer CNN, with each layer followed by Batch normalization and ReLU activation. The output of the CNN is aggregated into a feature vector via global average pooling, which is then fed to an LSTM and a single-layer MLP to generate logits.

The categorical cross-entropy loss is computed for 5 classes and the model with the highest mean intersection over union (mIOU) on the validation set is selected for further evaluation. Detailed model architecture and training settings are provided in the Appendix B.4. In PGA-DPS, the proportions of prior (deterministic) sampling ($Ps$) and active sampling ($As$) were 80% and 20%, respectively.

Three different DPS methods (DPS, A-DPS, and PGA-DPS) are implemented and compare to the conventional methods including uniform sampling, SRL-SOA (Self-Representation Learning with Sparse 1D-Operational Autoencoder) (Ahishali et al., 2022), and GSS (Greedy Spectral Selection) (Morales et al., 2021). In the uniform strategy, equidistant spectral bands are selected. In SRL-SOA, the polynomial order of the Taylor series expansion used to estimate the original signal using the autoencoder is set to 3. In the GSS approach, classification is performed on the center pixel of a 5 $\times$ 5 patch. For comparison, we also implemented the reference method 'All-bands', which does not use band selection and utilizes all 51 hyperspectral images.

### 4.5.3 RESULTS

Table 6 shows segmentation performance using three metrics: MPCA (mean per-class accuracy), mIOU (mean Intersection over Union), and mDICE (mean Sørensen–Dice coefficient). PGA-DPS outperformed all other methods, achieving performance comparable to that of using all 51 bands, while enabling over 10-fold acceleration. A-DPS produces less reliable segmentation maps, likely because the complexity of the segmentation makes joint optimization of the sampling strategy and multiple task models difficult. To preserve optimization stability, PGA-DPS limits active sampling at 20%. Example segmentation maps from subsampled bands selected by All-bands, PGA-DPS, A-DPS, DPS, and other band selection methods are shown in the Appendix (Fig. 4).

Table 6: Average segmentation results over 10 runs on the hold-out AeroRIT test set (3,127 patch samples) for band selection (using 5 bands, i.e. $r \approx 9.8\%$), compared to other approaches.

| | Band Selection Model | | | | | | |
| --- | --- | --- | --- | --- | --- | --- | --- |
| | All bands (Ref) | Uniform | SRL-SOA | GSS | DPS | A-DPS | PGA-DPS |
| MPCA (↑) | **88.36** | 80.59 | 83.68 | 77.91 | 85.75 | 65.55 | 86.37 |
| mIOU (↑) | **0.7037** | 0.5992 | 0.6160 | 0.6220 | 0.6703 | 0.5181 | 0.6752 |
| mDICE (↑) | **0.8063** | 0.6937 | 0.7476 | 0.7144 | 0.7732 | 0.6158 | 0.7803 |

## 5 CONCLUSION

We developed a scalable, stable active subsampling technique, called Prior-aware and Group-based Active DPS (PGA-DPS), that integrates context-guided group sampling and deterministic (fixed) subsampling based on training-data prior. PGA-DPS not only provides a theoretical analysis for the stable optimization of group sampling, enabled by DPS-top-$k$ rather than DPS-top-1, but also shows consistent and improved performance compared to conventional A-DPS. We demonstrated the effectiveness of PGA-DPS across three diverse tasks—classification, image reconstruction, and segmentation—using the MNIST, CIFAR-10, fastMRI knee, and AeroRIT datasets, outperforming all other sampling methods. Since top-k sampling and prior-aware deterministic sampling can be easily applied to various sampling tasks, PGA-DPS may serve as a powerful tool for improving performance in real-world applications where acquisition cost and inference efficiency are critical, such as CT, ultrasound, and radar systems.

## 6 LIMITATION & FUTURE WORK

Although PGA-DPS excels at generating subsampling strategies for diverse tasks, one possible limitation lies in selecting the hyperparameters associated with the proportion of fixed and active subsampling, denoted as $Ps$ and $As$, respectively. The optimal sampling portion differs depending on the downstream task, task model, and sampling ratio. We observed that an active sampling ratio ($As$) range of 20-30% provides the best trade-off between the benefit of top-k sampling and the total budget allocated to active sampling, which narrows the search space for optimal configurations.

To determine an appropriate prior-sampling ratio ($Ps$), we recommend using the performance discrepancy between DPS and A-DPS as an empirical indicator of the underlying Lipschitz characteristics of the task. Specifically, if DPS consistently outperforms A-DPS under a given configuration, this suggests that the optimization landscape exhibits a larger effective Lipschitz constant—implying that a higher Ps would be beneficial. Conversely, if A-DPS achieves superior performance, the landscape is likely smoother, in which case a smaller Ps is preferable.

While optimal hyperparameter (*Ps, As*) ranges are suggested, automatically tuning them (Bergstra et al., 2011; Feurer & Hutter, 2019; Bischl et al., 2023) could enhance the robustness of the proposed method. Future research may explore joint optimization of the active sampling hyperparameters (*Ps, As*), the subsampling trajectory, and the downstream task model.

We demonstrate that using a fixed temperature yields robust results (Appendix D); however, tuning or annealing the temperature could further enhance DPS-based methods. A more comprehensive investigation of temperature scheduling therefore represents a promising avenue for future work.

ACKNOWLEDGMENTS

This work was supported, in part, by the Institute of Information & Communications Technology Planning & Evaluation(IITP)-ITRC(Information Technology Research Center) grant funded by the Korea government(MSIT)(IITP-2026-RS-2024-00436857), and by the National Research Foundation of Korea(NRF) grant funded by the Korea government(MSIT)(NRF-2026-RS-2024-00338025).

REPRODUCIBILITY STATEMENT

The experimental settings and models used in our work are described in the Experimental Setup sections of each task. The task models and baseline approaches are explained in detail in the Task Model section. Additional implementation details are provided in the Appendix .

ETHICS STATEMENT

We used large language models (LLM) solely for line-level language editing, including grammar correction and minor phrasing refinements, during the preparation of this paper. LLMs were not involved in any technical aspects of the work.

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

APPENDIX

## A    PROOF

In this Appendix, we prove the Theorem 1 in Method. We provide a proof for the effect of DPS-top-1 sampling on the Lipschitzness of the loss.

**Theorem 1.** Let $f_1, f_2, \ldots, f_k$ be task models at $j^{th}$ iteration, where each function $f_j$ has a corresponding Lipschitz constants $L_j$ for $j = 1, 2, \ldots, k$. The In DPS-top-1, k task functions are required for each input sample $x_k$, whereas DPS-top-k defines a single task function $f_k$ over group of k samples. Lipschitzness of the loss function for each DPS-top-1 is as follows:

$$\|\text{Loss}_{\text{DPS-top-1}}(x)\|^2 = \|f(x^*) - f(x)\|^2 \le \prod_{r=1}^{k} L_r \|x^* - x\|^2, \quad f = f_k(f_{k-1}(\ldots(f_1(x_1))\ldots))$$

$$\|Loss_{\text{DPS-top-}k}(x)\|^2 = \|f_k(x^*) - f_k(x)\|^2 = \|f_k(x^*) - f_k(x_1, x_2, \ldots, x_k)\|^2 \le L_k \|x^* - x\|^2$$

Here, $x^* \in \mathbb{R}^N$ is the fully sampled input signal, $x \in \mathbb{R}^N$ is the sampled input signal (unsampled signal = 0), and $x_r$ is the $r^{th}$ sample. If the Lipschitz constant of task model at each iteration $L_j \ge 1$,

$$L_k \le \prod_{r=1}^{k} L_r \implies \sup_x \|\text{Loss}_{\text{DPS-top-}k}(x)\|^2 \le \sup_x \|\text{Loss}_{\text{DPS-top-1}}(x)\|^2$$

*Proof.* The Lipschitzness of the loss function for DPS-top-1 at first iteration is

$$\|f(x^*) - f_1(x_1)\|^2 \le L_1 \|x^* - x_1\|^2$$

The Lipschitzness of the loss function for DPS-top-1 at $j^{th}$ iteration is

$$\begin{aligned}
\|f(x_j^*) - f_j(x_j)\|^2 &\le L_j \|x_j^* - x_j\|^2 \\
&= L_j \|f_{j-1}(x_j^*) - f_{j-1}(x_{j-1})\|^2 \\
&\le L_j \times L_{j-1} \|x_{j-1}^* - x_{j-1}\|^2
\end{aligned}$$

By the principle of mathematical induction,

$$\|\text{Loss}_{\text{DPS-top-1}}(x)\|^2 = \|f(x^*) - f_k(x_k)\|^2 \le \prod_{r=1}^{k} L_r \|x^* - x_1\|^2$$

The Lipschitz constant is $\prod_{r=1}^{k} L_r$ for $\text{Loss}_{\text{DPS-top-1}}(x)$. On the other hand, the Lipschitzness of the loss function for DPS-top-$k$ is

$$\|Loss_{\text{DPS-top-}k}(x)\|^2 = \|f_k(x^*) - f_k(x)\|^2 \le L_k \|x^* - x\|^2$$

The Lipschitz constant of the loss function for DPS-top-$k$ is simply $L_k$. Therefore, if the Lipschitz constant of task model at each iteration $L_j \ge 1$,

$$L_k \le \prod_{r=1}^{k} L_r \implies \sup_x \|\text{Loss}_{\text{DPS-top-}k}(x)\|^2 \le \sup_x \|\text{Loss}_{\text{DPS-top-1}}(x)\|^2$$

## B    MODEL ARCHITECTURE AND TRAINING DETAILS

We adopt the same architectures and training strategies used in A-DPS for MNIST classification and MRI reconstruction tasks (Van Gorp et al., 2021). All experiments are run on a single NVIDIA RTX 3090 GPU.

### B.1 MNIST CLASSIFICATION

The classification network $f_\theta(\cdot)$ consists of 5-layer MLP with 784, 256, 128, 128, and 10 nodes, respectively. Each layer was followed by a leaky ReLU activation with a negative slope of 0.2 and the last layer followed by softmax activation that output probabilities of class. A dropout rate of 30% is also applied to the first three layers.

The sampling network $g(\cdot)$ consists of an LSTM (Long Short-Term Memory) with one hidden layer of size 128, followed by two MLPs with 256 and 784 nodes, respectively. A leaky ReLU activation with a negative slope of 0.2 and a dropout rate of 30% are applied after the first MLP.

The categorical cross-entropy loss was minimized using the Adam optimizer (Kingma & Ba, 2014) with the learning rate of $2 \times 10^{-4}$ ($\beta_1 = 0.9$, $\beta_2 = 0.999$, and $\epsilon = 10^{-7}$). In addition, the trainable logits, of size 784 (nn.Parameters()), are also updated with a learning rate of $2 \times 10^{-3}$ for DPS and PGA-DPS. The optimization was run for 100 epochs with a batch size of 256.

### B.2 CIFAR-10 CLASSIFICATION

The classification network $f_\theta(\cdot)$ consists of four convolutional layers with channel sizes of 32, 64, and 128, respectively, and a kernel size of $3 \times 3$. Each layer is followed by batch normalization, a ReLU activation function, and a $2 \times 2$ max pooling operation. The resulting 128-channel feature map is then flatten and fed into an MLP composed of Linear(2048,256)-ReLU-Dropout(0.5)-Linear(256,128)-ReLu-Linear(128,10).

The sampling network $g(\cdot)$ consists of an LSTM (Long Short-Term Memory) with one hidden layer of size 128, followed by two MLPs with 256 and 1024 nodes, respectively. A leaky ReLU activation with a negative slope of 0.2 and a dropout rate of 30% are applied after the first MLP.

The categorical cross-entropy loss was minimized using the Adam optimizer with the learning rate of $2 \times 10^{-4}$. In addition, the trainable logits, of size 1024 (nn.Parameters()), are also updated with a learning rate of $2 \times 10^{-3}$ for DPS and PGA-DPS. The optimization was run for 50 epochs with a batch size of 128.

### B.3 MRI RECONSTRUCTION

The proximal neural network $\mathcal{P}_\psi$ consists of four convolutional layers with channel size of 16,16,16, and 1, respectively, and a kernel size of $3 \times 3$. Each layer is followed by a ReLU activation function except for the final layer. For the image prior regularizer ($\psi$), a single $3 \times 3$ convolutional layer is used. The proximal gradient method is unrolled for three iterations (k=3).

The adversarial loss (Ledig et al., 2017), combined with a mean square error (MSE), is used to reconstruct visually plausible MR images, by training a discriminator network that distinguishes the real and reconstructed images. In addition, the discriminator feature loss between the real and reconstructed images was employed.

The sampling network $g^k(\cdot)$ consists of a sequence of layers: Conv2d(1,16), ReLU, Conv2d(16,32), ReLu, Conv2d(32,64), ReLU, Average Pooling, Flatten, LSTM(64), and a single-layer MLP(64,208). All convolutional layers have a kernel size of $3 \times 3$. The last MLP converts the output of the LSTM (encoded context) to the logits of size 208.

The discriminator network is consists of a sequence of layers: Conv2d(1,64), LeakyReLU(0.2), Conv2d(64,64), LeakyReLU(0.2), Conv2d(64,64), LeakyReLU(0.2), Average Pooling, Dropout(0.4), a single-layer MLP (64,1), and Sigmoid. All convolutional layers have a kernel size of $3 \times 3$ and a stride of 2.

The total loss is computed as a weighted sum of three losses: mean squared error (MSE), adversarial loss, and the discriminator feature loss, with respective weights of 1, $5 \times 10^{-6}$, and $10^{-7}$.

The network is updated using the Adam optimizer (Kingma & Ba, 2014) with the learning rate of $2 \times 10^{-4}$ ($\beta_1 = 0.9$, $\beta_2 = 0.999$, and $\epsilon = 1e - 7$). In addition, the trainable logits, of size 208 (nn.Parameters()), are also updated with a learning rate of $2 \times 10^{-3}$ for DPS and PGA-DPS. The optimization was run for 10 epochs with a batch size of 2.

### B.4 HYPERSPECTRAL IMAGE SEGMENTATION

For the segmentation network $f_\theta(\cdot)$, we use a residual U-Net consisting of 6 ResNet blocks, as proposed in the pix2pix (Isola et al., 2017; Zhu et al., 2017). The input channel is set to 51, corresponding to the total number of hyperspectral bands, and the output channel is set to 6, corresponding to the number of classes including the undefined class. The number of the filters in the last convolution layer is set to 64, and tow downsampling operations are applied.

The sampling network $g^k(\cdot)$ consists of a sequence of layers: Conv2d(6,32), BN, ReLU, Conv2d(32,64), BN, ReLu, Conv2d(64,128), BN, ReLU, Conv2d(128,256), BN, ReLU, Average Pooling, Flatten, LSTM(256), and a single-layer MLP(256,51), where BN denotes bach normalization. All convolutional layers have a kernel size of $3 \times 3$. The last MLP converts the output of the LSTM (encoded context) to the logits of size 51.

In PGA-DPS, the proportions of prior (deterministic) sampling ($N$) and active sampling ($M$) were 80% and 20%, respectively, resulting in two group sampling iterations.

The network is updated using the Adam optimizer (Kingma & Ba, 2014) with the learning rate of $10^{-4}$ ($\beta_1 = 0.9$, $\beta_2 = 0.999$, and $\epsilon = 1e - 7$), while the learning rate of $2 \times 10^{-4}$ is used for DPS and GSS. In addition, the trainable logits, of size 51 (nn.Parameters()), are also updated with a learning rate of 1 for DPS and PGA-DPS. The optimization was run for 60 epochs with a batch size of 100.

## C RESULTS

### C.1 MNIST CLASSIFICATION

We evaluated multiple $Ps$ values under an $As$ of 20% and multiple $As$ values under a $Ps$ of 60%. Table 7 shows that PA-DPS outperforms the baselines in almost all configurations, although the optimal configuration shifts slightly across sampling ratios.

Table 7: The classification accuracy on the MNIST test set (10,000 samples) for various subsampling ratios ($r$) using various configurations of PGA-DPS ($Ps,As$). Each sampling strategy was repeated across six independent runs.

| Sampling Model | Sampling ratio: r (%) | | | | | | | | |
| --- | --- | --- | --- | --- | --- | --- | --- | --- | --- |
| | 1 | 2 | 3 | 4 | 5 | 6 | 7 | 8 | 100 |
| DPS | 63.5 | 83.6 | 91.5 | 95.2 | 96.4 | 97.1 | 97.3 | 97.6 | 98.2 |
| A-DPS | 64.6 | 85.2 | 92.2 | 95.3 | 96.1 | 96.6 | 97.1 | 97.2 | - |
| PGA-DPS (80,20) | 69.1 | 86.0 | 93.9 | 95.8 | 96.8 | 97.3 | 97.6 | 97.9 | - |
| PGA-DPS (60,20) | 68.8 | 86.9 | 93.9 | 95.8 | 96.7 | 97.2 | 97.5 | 97.7 | - |
| PGA-DPS (40,20) | 67.8 | 86.8 | 93.6 | 95.9 | 96.7 | 97.3 | 97.4 | 97.6 | - |
| PGA-DPS (20,20) | 66.8 | 85.9 | 93.2 | 95.6 | 96.4 | 97.0 | 97.2 | 97.5 | - |
| PGA-DPS (60,5) | 68.8 | 87.1 | 93.7 | 95.8 | 96.5 | 97.0 | 97.2 | 97.6 | - |
| PGA-DPS (60,10) | 68.8 | 87.1 | 93.8 | 95.7 | 96.7 | 97.2 | 97.4 | 97.7 | - |
| PGA-DPS (60,15) | 68.8 | 87.6 | 94.2 | 95.8 | 96.6 | 97.2 | 97.5 | 97.7 | - |
| PGA-DPS (60,25) | 68.8 | 87.0 | 93.8 | 95.8 | 96.7 | 97.2 | 97.5 | 97.7 | - |
| PGA-DPS (60,30) | 68.3 | 87.0 | 93.9 | 95.7 | 96.7 | 97.2 | 97.5 | 97.7 | - |

### C.2 MRI RECONSTRUCTION

Figure 3 shows example reconstruction from masks generated by PGA-DPS, A-DPS, DPS, and other sampling methods. Notably, PGA-DPS selects samples farthest from the DC, capturing fine-grained details. PGA-DPS exhibits the farthest average mask distance from the DC (11.8), versas 10.2 (A-DPS), 8.8 (DPS), 10.8 (LOUPE), 8.1 (Greedy Prox.), and 11.3 (VDS). Despite prioritizing high-frequency components, PGA-DPS also samples adequately near DC, demonstrating its superior reconstruction performance with higher SSIM values.

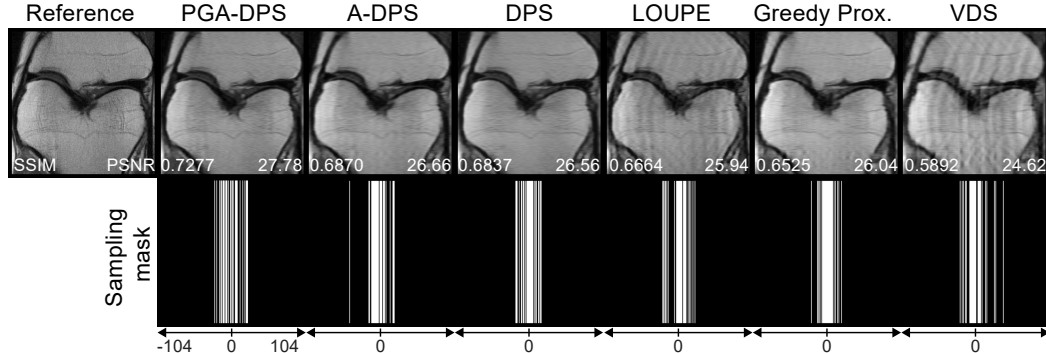

Figure 3: Three examples of subsampling masks ($r = 12.5\%$) and the corresponding reconstructed MR images obtained from PGA-DPS, A-DPS, and DPS. Results are compared against reference images reconstructed from fully sampled k-space data.

To evaluate the robustness and general applicability of the proposed method, we tested whether the selected optimal configuration of (Ps, As) = (30%, 30%), identified from analysis with k-space lines M = 26, generalizes well across different sampling ratios by conducting experiments with varying values of M, i.e., k-space lines M = 15, 20, 30, and 35, corresponding to acceleration rates r = 7.2%, 9.6%, 14.4%, and 16.8%, respectively. As shown in Table 8, PGA-DPS outperforms DPS and A-DPS across various numbers of k-space lines, with a particularly large margin when the number of k-space lines is small.

Table 8: Average results over 5 runs on the hold-out test set of size $208 \times 208$ pixels for various acceleration factors ($r = M/N$, $N = 208$), compared to the DPS and A-DPS methods.

| Method | Metric | M = 15 | 20 | 30 | 35 |
|---|---|---|---|---|---|
| DPS | NMSE | 0.0517 | 0.0448 | 0.0354 | 0.0325 |
| | PSNR | 23.9 | 24.7 | 26.0 | 26.5 |
| | SSIM | 0.476 | 0.528 | 0.623 | 0.660 |
| A-DPS | NMSE | 0.0571 | 0.0497 | 0.0360 | 0.0333 |
| | PSNR | 23.3 | 24.2 | 25.9 | 26.4 |
| | SSIM | 0.444 | 0.513 | 0.613 | 0.647 |
| PGA-DPS (30%, 30%) | NMSE | **0.0472** | **0.0402** | **0.0325** | **0.0304** |
| | PSNR | **24.4** | **25.3** | **26.4** | **26.9** |
| | SSIM | **0.505** | **0.568** | **0.657** | **0.690** |

### C.3 HYPERSPECTRAL IMAGE SEGMENTATION

Figure 4 shows example segmentation maps from subsampled bands selected by All-bands, PGA-DPS, A-DPS, DPS, and other band selection methods.

## D ABLATION STUDY ON TEMPERATURE IN DPS ARCHITECTURE

We fixed the temperature to 2 during training for DPS, A-DPS, and PGA-DPS. However, tuning the relaxation parameter is critical for balancing the bias-variance trade-off of the gradient estimator and may lead to further improvements.

To assess this, we conducted additional MR reconstruction experiments using various fixed temperature values. As shown in Table 9, the reconstruction performance was largely insensitive to temperature values. Nonetheless, a fixed temperature of 5 yielded the best overall results, suggesting that temperature optimization could provide marginal additional improvements.

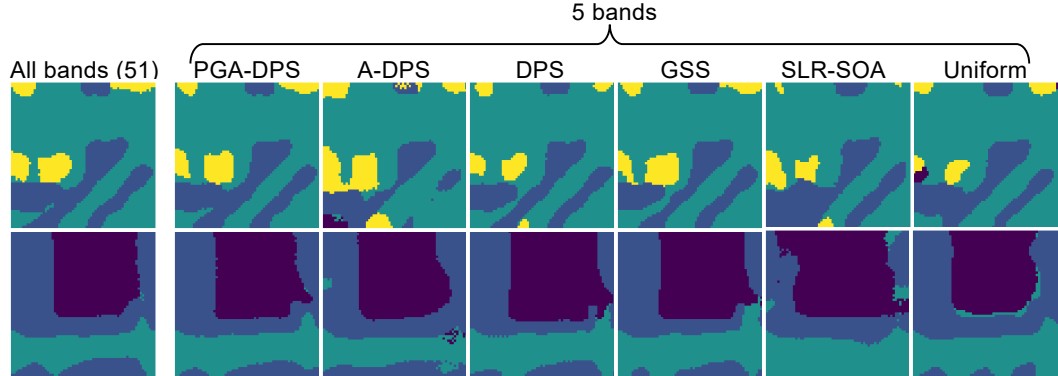

Figure 4: Two example segmentation maps estimated using 5 bands selected from PGA-DPS, A-DPS, DPS, and other band selection methods ($r \approx 9.8\%$). Results are compared against reference segmentation map estimated using all 51 spectral bands.

Table 9: Average results over 5 runs on the hold-out test set ($208 \times 208$ pixels) for an acceleration factor of 8 (M = 26, r = 12.5%), evaluated across various fixed temperatures using PGA-DPS.

|  | $\tau = 0.5$ | 1.0 | 2.0 | 3.0 | 5.0 | 10.0 |
|---|---|---|---|---|---|---|
| NMSE | 0.0354 | 0.0354 | 0.0350 | 0.0355 | **0.0349** | 0.0353 |
| PSNR | 26.00 | 26.00 | 26.05 | 25.97 | **26.08** | 26.02 |
| SSIM | 0.620 | 0.624 | 0.625 | 0.619 | **0.627** | 0.625 |

We also evaluated temperature annealing, defined as, $\tau = max(2.0, T \times e^{-\gamma n})$, where n denotes the training epoch. As shown in Table 10, different annealing schedules produced noticeably different performance. Consequently, using a fixed temperature provides sufficiently robust and stable results in our setting, consistent with what has been found to work well in practice (Huijben et al., 2020a; Van Gorp et al., 2021). Nevertheless, a more comprehensive investigation of temperature annealing within DPS remains a promising direction for future work.

Table 10: Average results over 5 runs on the hold-out test set ($208 \times 208$ pixels) for an acceleration factor of 8 (M = 26, r = 12.5%), evaluated across various temperature annealing strategies ($\tau = max(2.0, T \times e^{-\gamma n})$) using PGA-DPS.

|  |  | $\gamma = 0.1$ | 0.2 | 0.3 | 0.4 | 0.5 |
|---|---|---|---|---|---|---|
| | NMSE | 0.0378 | 0.0362 | 0.0381 | 0.0364 | 0.0367 |
| T=5 | PSNR | 25.7 | 25.9 | 25.6 | 25.9 | 25.8 |
| | SSIM | 0.582 | 0.606 | 0.584 | 0.604 | 0.599 |
| | NMSE | 0.0389 | **0.0353** | 0.0379 | 0.0361 | 0.0378 |
| T=10 | PSNR | 25.6 | **26.00** | 25.68 | 25.9 | 25.7 |
| | SSIM | 0.581 | **0.621** | 0.586 | 0.607 | 0.585 |

