# OpenReview forum: "Prior-aware and Context-guided Group Sampling for Active Probabilistic Subsampling"
_ICLR.cc/2026/Conference — ICLR 2026 Poster_

### Official Review · Reviewer_9EHq · 2025-10-25

**Soundness:** 3
**Presentation:** 3
**Contribution:** 2
**Rating:** 4
**Confidence:** 4

**Summary:**

This paper proposes Prior-aware and Context-guided Group-based Active Deep Probabilistic Subsampling (PGA-DPS), which enhances Active Deep Probabilistic Subsampling (A-DPS) by incorporating two key innovations: (1) integrating deterministic prior-informed sampling patterns derived from training data statistics with instance-specific active sampling; (2) introducing DPS-top-k group sampling to replace sequential top-1 sampling. The authors provide theoretical analysis showing that group sampling achieves smaller effective Lipschitz constants, leading to smoother optimization landscapes. Experiments on MNIST classification, fastMRI reconstruction, and hyperspectral segmentation demonstrate that PGA-DPS outperforms existing sampling methods across all evaluation metrics.

**Strengths:**

1. The dual sampling strategy combining deterministic priors with adaptive active sampling is well-motivated and novel. While group sampling is not entirely new, its application and theoretical analysis in this context show originality.

2. The method is technically sound with solid implementation. The theoretical analysis (Theorem 1 on Lipschitz constants) provides valuable insights into optimization stability. The experimental design covers three diverse domains, demonstrating method generality.

3. The paper is well-structured with clear method description and reasonable figure presentations.

**Weaknesses:**

1. Questionable practical relevance: The MNIST pixel-level sampling has limited real-world significance as pixel acquisition costs are identical and downsampling doesn't provide substantial practical benefits. For MRI and HSI tasks, while acceleration is mentioned, detailed time cost analysis and end-to-end system evaluation are missing.

2. Incomplete experimental evaluation: All experiments lack 100% data sampling baselines, making it impossible to assess performance degradation at different sampling rates. This significantly undermines the assessment of the method's practical value.

3. Limited dataset scale: MNIST is too simple to validate method scalability. With sufficient computational resources, evaluation on CIFAR-10/100 or larger datasets would strengthen the claims. While fastMRI and AeroRIT are more realistic, their scales remain relatively limited.

**Questions:**

1. Annotation dependency of prior sampling: Prior sampling requires complete training data annotations. How is this handled in annotation-scarce scenarios? How to update prior sampling patterns when new data continuously arrives?

2. Hyperparameter sensitivity: The (Ps, As) ratios vary significantly across tasks. Is there an automatic selection strategy?

Others can be seen in Weaknesses.

---

> ### Author Response · Authors · 2025-11-21
> **Response to Reviewer 9EHq (1/3)**
>
> Thank you for recognizing the strengths of our work. We also appreciate your thoughtful feedback, which highlights areas where further clarification can enhance the paper. Below, we address your comments and questions, aiming to fully resolve your concerns.
>
> >**Q1. Annotation dependency of prior sampling: Prior sampling requires complete training data annotations. How is this handled in annotation-scarce scenarios? How to update prior sampling patterns when new data continuously arrives?**
>
> Prior sampling in PGA-DPS is designed to determine where to sample within the _Ps_ portion by learning a deterministic, data-driven sampling pattern. Importantly, this prior pattern does not require full annotations for every newly arriving dataset. The deterministic mask is learned once using the available annotated data, and thereafter acts as a static prior during deployment.
>
> In annotation-scarce scenarios, PGA-DPS simply learns the deterministic mask from the annotated subset and then applies it universally across future acquisitions, similar to how static masks are routinely used in classical compressed sensing. Thus, the annotation requirement is one-time rather than continuous.
>
> When new data arrives sequentially, PGA-DPS can first populate the _Ps_ portion using the learned deterministic mask. Once those positions are filled, active sampling begins over the remaining _As_	portion using DPS-top-𝑘. Likewise, if the _As_ region has not yet been reached, PGA-DPS can defer updating the sampling locations (pattern) until sufficient data has accumulated.
>
> >**Q2. Hyperparameter sensitivity: The (Ps, As) ratios vary significantly across tasks. Is there an automatic selection strategy?**
>
> This is an important question. As the reviewer noted, while PGA-DPS excels at generating subsampling strategies across diverse tasks, one potential limitation lies in selecting the hyperparameters that determine the proportion of fixed and active subsampling, denoted as _Ps_ and _As_, respectively.
>
> In our method, the optimal sampling portion varies slightly depending on the downstream task, task model, and the sampling ratio. We observed that an active sampling ratio (_As_) range of 20-30\% provides the best trade-off between the benefit of top-k sampling and the total budget allocated to active sampling. This observation substantially narrows the search space for selecting optimal _Ps_ and _As_ combinations. Additional experiments on the CIFAR-10 dataset also confirmed that _As_ within this range works well, as shown in Table 2 of the revised manuscript.
>
> To determine an appropriate prior-sampling ratio (Ps), we recommend using the performance discrepancy between DPS and A-DPS as an empirical indicator of the underlying Lipschitz characteristics of the task. Specifically, if DPS consistently outperforms A-DPS under a given configuration, this suggests that the objective landscape exhibits a larger effective Lipschitz constant—implying that a higher Ps would be beneficial. Conversely, if A-DPS achieves superior performance, the landscape is likely smoother (i.e., associated with a smaller Lipschitz constant), in which case a smaller Ps is preferable. For example, in the MRI reconstruction task, A-DPS outperforms DPS, and thus we used a small _Ps_ of 30 \%. In contrast, in the HSI segmentation task, DPS outperforms A-DPS, so we opted for a large _Ps_ of 80%.
>
> While we suggest a strategy for choosing optimal hyperparameter of (_Ps, As_), automatically tuning them (Bergstra et al., 2011; Feurer & Hutter, 2019; Bischl et al., 2023) could further improve the robustness of the proposed method. Future work may explore joint optimization of the active sampling hyperparameters (_Ps, As_), the subsampling trajectory, and the downstream task model. We have added this discussion to the “LIMITATION & FUTURE WORK” section in the revised version of the main paper.

---

> ### Author Response · Authors · 2025-11-21
> **Response to Reviewer 9EHq (2/3)**
>
> **Weaknesses:**
>
> >**W1. Questionable practical relevance: The MNIST pixel-level sampling has limited real-world significance as pixel acquisition costs are identical and downsampling doesn't provide substantial practical benefits. For MRI and HSI tasks, while acceleration is mentioned, detailed time cost analysis and end-to-end system evaluation are missing.**
>
> As the reviewer noted, pixel-level sampling on MNIST has limited real-world relevance. We included the MNIST experiments primarily to demonstrate that our method generalizes across heterogeneous tasks rather than restricted to a specific domain.
>
> In contrast, acceleration is critical in MRI and HIS applications because data acquisition is costly and the acquisition time grows approximately linearly with number of sampled lines or spectral bands. For example, the AeroRIT scene was collected using two airborne hyperspectral camera systems flown over the Rochester Institute of Technology’s university campus, with data acquisition spanning several hours (Sun & Du, 2019). Similarly, the FastMRI data were acquired with 2D turbo spin echo (TSE) sequences, which typically requires several minutes for a whole-brain scan. For quantitative MRI techniques based on TSE, such as saturation-transfer MR fingerprinting, scan times can extend to tens of minutes (10.1002/mrm.30532). Thus, reducing acquisition time is essential for limiting motion artifacts and improving patient comfort.
>
> >**W2. Incomplete experimental evaluation: All experiments lack 100\% data sampling baselines, making it impossible to assess performance degradation at different sampling rates.**
>
> Thank you for pointing out this ambiguity. To clarify, we have added the 100\% data sampling baseline for the MNIST classification task in Table 1. For the HSI segmentation task, the corresponding baseline is already provided in Table 6 as “All bands (Ref)”.
>
> For MRI reconstruction task, the goal is to recover the fully-sampled image from subsampled k-space data. Consequently, a quantitative metric for comparison against a ‘fully sampled’ baseline is not applicable. Instead, we show the fully sampled reconstruction in Figure 3 of the Appendix as the ‘Reference’ image.

---

> ### Author Response · Authors · 2025-11-21
> **Response to Reviewer 9EHq (3/3)**
>
> >**W3. Limited dataset scale: MNIST is too simple to validate method scalability. With sufficient computational resources, evaluation on CIFAR-10/100 or larger datasets would strengthen the claims. While fastMRI and AeroRIT are more realistic, their scales remain relatively limited.**
>
> To further strengthen our work, we conducted an additional experiment on the CIFAR-10 dataset. The proposed PGA-DPS outperformed both A-DPS and DPS methods. The full experimental setup and results are provided below and have also been incorporated into the revised main manuscript (See section 4.2. CIFAR-10 classification).
>
> **Experimental Settings.** We evaluated pixel-level subsampling for classification using the CIFAR-10 dataset (Krizhevsky et al., 2009), which contains 60,000 color images of size 32 x 32 across 10 classes. Of these, 50,000 images were used for training and validation, and 10,000 for testing. The training split was further divided into training and validation set at a 9:1 ratio. We compared the proposed PGA-DPS, DPS and A-DPS methods across sampling ratios ranging from 2\% to 20\% in increment of 2\%.
>
> **Task model.** For the classification model $f_{\theta}(\cdot)$, we employed a four-layer convolutional neural network (CNN) followed by three MLP layers. The CIFAR-10 images are first masked according to the subsampling pattern and then fed into the network. We employed the same-sampling network architecture and the same cross-entropy loss function as in the MNIST classification experiments. The detailed model architecture and training settings are provided in the Appendix B.2. In PGA-DPS, the proportions for prior sampling and active sampling are fixed to 10\% and 20\%, respectively.
>
> **Result.**
> Table R3.1 shows that the proposed PGA-DPS outperforms both DPS and A-DPS across all sampling ratios. Because CIFAR-10 classification is substantially more challenging than MNIST, the performance gains are larger across the entire range. Unlike the MNIST classification task, A-DPS exhibits consistently strong performance relative to DPS for all ratios. This behavior is likely attributable to the lower Lipschitz constant of CNNs compared to MLPs used in the MNIST experiments (Shi et al.,2022), which makes the optimization landscape easier to navigate. However, beyond a certain number of samples ( r = 14 \%), the classification accuracy of A-DPS decreased, likely due to an inflated Lipschitz constant, similar to what was observed in the MNIST classification task.
>
>
> **Table R3.1.** The classification accuracy on the CIFAR-10 test set (10,000 samples) for various subsampling ratios (r). Each sampling strategy was repeated across six independent runs.
> | r (%)    | 2    | 4    | 6    | 8    | 10   | 12   | 14   | 16   | 18   | 20   | 100  |
> |----------|------|------|------|------|------|------|------|------|------|------|------|
> | DPS      | 42.2 | 44.5 | 43.4 | 44.1 | 46.1 | 46.5 | 49.2 | 50.5 | 52.1 | 54.3 | 96.3 |
> | A-DPS    | 52.9 | 61.0 | 65.8 | 67.4 | 70.4 | 69.6 | 70.7 | 69.5 | 70.0 | 68.3 | -    |
> | PGA-DPS  | 54.3 | 63.2 | 68.6 | 70.8 | 74.7 | 78.1 | 78.6 | 79.8 | 81.3 | 82.7 | -    |

---

> > ### Comment · Reviewer_9EHq · 2025-11-26
> >
> > Thank you for the comprehensive rebuttal. The inclusion of the CIFAR-10 experiments validates the scalability of the proposed method and resolves my main concerns. I also appreciate the clarification on the practical value in MRI/HSI scenarios. Consequently, I am raising my score.

---

> ### Author Response · Authors · 2025-11-27
>
> We greatly appreciate your recognition of the key contributions and strengths of our work.
>
> If you have any further questions or suggestions, please feel free to reach out. We are fully committed to addressing any concerns or providing additional clarifications.
>
> Thank you once again for your constructive and thoughtful review.
>
>
> Best regards,
>
> The Authors

---

### Official Review · Reviewer_yF5c · 2025-10-31

**Soundness:** 3
**Presentation:** 2
**Contribution:** 2
**Rating:** 4
**Confidence:** 3

**Summary:**

The main idea of this paper is to augment Deep Probabilisitic Subsampling (DPS) for task adaptive subsampling with Active Deep Probabilistic Subsampling method (A-DPS). The main idea behind the DPS method is to use Gumbel-softmax trick for back-propagation to estimate the gradients.

In this paper, the proposed method first performs the deterministic subsampling from the training data and then performs the A-DPS method for contextually selecting more samples. Also, another modification proposed is to use DPS-K for sampling instead of DPS-1 in the A-DPS step.

Empirical results on MNIST and MRI datasets show significant improvements over existing baselines.

**Strengths:**

1. Empirical results show that the usefulness of the proposed method. Significant gains on MNIST and MRI reconstruction dataset.

2. The method is benchmarked against a wide range of alternatives—including greedy algorithms, learned masks (LOUPE), RL methods, and spectral selection approaches proposed for other domains.

**Weaknesses:**

1. The core algorithm is a combination of already published ideas: deterministic mask learning (training-set prior as used in DPS and LOUPE), sequential active sampling (A-DPS), and top-k/group sampling (DPS-top-k, Gumbel-top-k) in a straightforward manner.

2. The tuning of how many samples come from the prior mask versus active sampling (Ps/As) is hand-picked for different tasks, and the contributions are hyperparameter-dependent.

3. The prior sampling part is not clear, and the key details are hidden away in appendix.

**Questions:**

1. How does this method compare against Attention guided methods [1]?

2. Expand on section 3.3.  Also, is Theorem 1 a known result or new contribution?

3. How sensitive is the method to the fixed temperature parameter

4. Can the hyperparameters P_s and A_s (proportions of prior and active sampling) be learned during training instead of fixed?

5. How does the prior-informed deterministic mask compare to purely learned or data-driven static masks in terms of interpretability and robustness?

6. How does the choice of task model (classification vs. reconstruction) affect the optimal proportion of prior vs. active samples?


References:

[1] Shankaranarayana, Sharath M., et al. "Deep Attention-guided Adaptive Subsampling." arXiv preprint arXiv:2510.12376 (2025).

Typos:

Line 017: impedes -> impede
Line 280: accrucay -> accuracy
Line 479: DSP-1 -> DPS-1

---

> ### Author Response · Authors · 2025-11-21
> **Response to Reviewer yF5c (1/3)**
>
> Thank you for recognizing the strengths of our work. We also appreciate your thoughtful feedback, which highlights areas where further clarification can enhance the paper. Below, we address your comments and questions, aiming to fully resolve your concerns.
>
> >**Q1. How does this method compare against Attention guided methods [1]?**
>
> The key difference is that our method (GPA-DPS) performs active probabilistic subsampling in the measurement domain (e.g., k-space lines, spectral bands, pixels), aiming to reduce acquisition cost. In contrast, Deep Attention-guided Adaptive Subsampling selects regions within already-acquired images, discarding slices or frames that are unnecessary for classification, thereby reducing computational cost rather than acquisition cost.
>
> Thus, GPA-DPS optimizes the sampling process itself—combining prior and active sampling and jointly optimizing it with the downstream task—whereas the attention-guided method is essentially an attention-based input trimming mechanism and does not address measurement-domain subsampling.
>
> Although the two approaches differ conceptually, a direct comparison with this recently released contemporary work---an attention-guided adaptive subsampling method would indeed be valuable. We are committed to performing this comparison as soon as the official code becomes publicly available and will incorporate the results in a subsequent revision of our work.
>
> >**Q2. Expand on section 3.3. Also, is Theorem 1 a known result or new contribution?**
>
> Theorem 1 offers a novel theoretical insight specific to our proposed GPA-DPS framework. Our theorem is the first to formally analyze group-wise active probabilistic sampling and show that grouping yields a smoother optimization objective under the DPS framework.
>
> We provide a complete proof in the Appendix demonstrating this effect, and the predicted optimization benefits are consistently supported by our empirical results across multiple tasks. These findings can directly address the reviewer’s concern that “the core algorithm might simply combine previously published ideas (Weakness)”. While our method builds upon components from existing approaches, we contribute a new theoretical analysis that establishes the superiority of top-k sampling over top-1 sampling in DPS, further supported by consistent empirical evidence.

---

> ### Author Response · Authors · 2025-11-21
> **Response to Reviewer yF5c (2/3)**
>
> >**Q3. How sensitive is the method to the fixed temperature parameter**
>
> We fixed the temperature to 2 during training for DPS, A-DPS, and PGA-DPS. However, tuning the relaxation parameter is critical for balancing the bias-variance trade-off of the gradient estimator and may lead to further improvements.
>
> To assess this, we conducted additional MR reconstruction experiments using various fixed temperature values. As shown in Table R2.1, the reconstruction performance was largely insensitive to temperature values. Nonetheless, a fixed temperature of 5 yielded the best overall results, suggesting that temperature optimization could provide marginal additional improvements.
>
> We also evaluated temperature annealing, defined as,
> $\tau = \max(2.0, T \times e^{-\gamma n})$,
> where n denotes the training epoch. As shown in Table R2.2, different annealing schedules produced noticeably different performance. Consequently, using a fixed temperature provides sufficiently robust and stable results in our setting, consistent with what has been found to work well in practice (Huijben et al., 2020a; Van Gorp et al., 2021). Nevertheless, a more comprehensive investigation of temperature annealing within DPS remains a promising direction for future work. We have incorporated these new results into the Appendix and addressed them in Discussion section of the revised manuscript.
>
> **Table R2.1.** Average results over 5 runs on the hold-out test set of size 208 × 208 pixels for various acceleration factors (_M_ = 26, _r_ = 12.5 \%), evaluated across various fixed temperatures ($\tau$) using PGA-DPS.
> | $\tau$ = | 0.5   | 1.0   | 2.0   | 3.0   | 5.0   | 10.0  |
> |----------------|-------|-------|-------|-------|-------|-------|
> | NMSE           | 0.0354| 0.0354| 0.0350| 0.0355| **0.0349**| 0.0353|
> | PSNR           | 26.0  | 26.00 | 26.05 | 25.97 | **26.08** | 26.02 |
> | SSIM           | 0.620 | 0.624 | 0.625 | 0.619 | **0.627** | 0.625 |
>
> **Table R2.2.** Average results over 5 runs on the hold-out test set of size 208 × 208 pixels for various acceleration factors (_M_ = 26, _r_ = 12.5 \%), evaluated across various temperature annealing strategies ($\tau = \max(2.0, T \times e^{-\gamma n})$) using PGA-DPS.
> | |$\gamma $ =  | 0.1    | 0.2    | 0.3    | 0.4    | 0.5    |
> |-----|--------|--------|--------|--------|--------|--------|
> | T = 5   | NMSE   | 0.0378 | 0.0362 | 0.0381 | 0.0364 | 0.0367 |
> |     | PSNR   | 25.7   | 25.9   | 25.6   | 25.9   | 25.8   |
> |     | SSIM   | 0.582  | 0.606  | 0.584  | 0.604  | 0.599  |
> | T  = 10  | NMSE   | 0.0389 | **0.0353** | 0.0379 | 0.0361 | 0.0378 |
> |     | PSNR   | 25.6   | **26.00**  | 25.68 | 25.9   | 25.7   |
> |     | SSIM   | 0.581  | **0.621**  | 0.586 | 0.607  | 0.585  |

---

> ### Author Response · Authors · 2025-11-21
> **Response to Reviewer yF5c (3/3)**
>
> >**Q4. Can the hyperparameters P_s and A_s (proportions of prior and active sampling) be learned during training instead of fixed?**
>
> >**Q6. How does the choice of task model (classification vs. reconstruction) affect the optimal proportion of prior vs. active samples?**
>
> These are important questions. As the reviewer commented, while PGA-DPS excels at generating subsampling strategies across diverse tasks, one potential limitation lies in selecting the hyperparameters that determine the proportion of fixed and active subsampling, denoted as _Ps_ and _As_, respectively.
>
> In our method, the optimal sampling portion varies slightly depending on the downstream task, task model, and the sampling ratio. We observed that an active sampling ratio (_As_) range of 20-30\% provides the best trade-off between the benefit of top-k sampling and the total budget allocated to active sampling. This observation substantially narrows the search space for selecting optimal _Ps_ and _As_ combinations. Additional experiments on the CIFAR-10 dataset also confirmed that _As_ within this range works well, as shown in Table 2 of the revised manuscript (See Section 4.2. CIFAR-10 Classification).
>
> To determine an appropriate prior-sampling ratio (Ps), we recommend using the performance discrepancy between DPS and A-DPS as an empirical indicator of the underlying Lipschitz characteristics of the task. Specifically, if DPS consistently outperforms A-DPS under a given configuration, this suggests that the objective landscape exhibits a larger effective Lipschitz constant—implying that a higher Ps would be beneficial. Conversely, if A-DPS achieves superior performance, the landscape is likely smoother (i.e., associated with a smaller Lipschitz constant), in which case a smaller Ps is preferable. For example, in the MRI reconstruction task, A-DPS outperforms DPS, and thus we used a small _Ps_ of 30 \%. In contrast, in the HSI segmentation task, DPS outperforms A-DPS, so we opted for a large _Ps_ of 80%.
>
> While we suggest a strategy for choosing optimal hyperparameter of (_Ps, As_), automatically tuning them (Bergstra et al., 2011; Feurer & Hutter, 2019; Bischl et al., 2023) could further improve the robustness of the proposed method. Future work may explore joint optimization of the active sampling hyperparameters (_Ps, As_), the subsampling trajectory, and the downstream task model. We have added this discussion to the “LIMITATION & FUTURE WORK” section in the revised version of the main paper.
>
>
>
> >**Q5. How does the prior-informed deterministic mask compare to purely learned or data-driven static masks in terms of interpretability and robustness?**
>
> Before discussing interpretability and robustness, it is useful to brefily revisit the proposed PGA-DPS framework. The prior-informed deterministic mask can be viewed as a data-driven static mask. In our implementation, we apply the conventional DPS strategy to the _Ps_ portion of the target samples. This clarification addresses the reviewer’s concern that “the prior sampling part is not clear (Weakness)”
>
> To enable active sampling, PGA-DPS integrates this deterministic prior with an active selection mechanism. Active sampling is performed using DPS-top-k on the _As_ portion, in contrast to A-DPS, which applies DPS-top-1 to all samples. Additional experiments using multiple random seeds, reported as averaged results, further demonstrate the robustness of the proposed approach.
>
> The interpretability of each method (DPS, A-DPS, and PGA-DPS) is partially discussed in Section 4.2.3. DPS predominantly selects on DC (low-frequency) lines, capturing global structural information. In contrast, PGA-DPS and A-DPS tend to select more peripheral k-space regions, enabling recovery of richer spatial details. Notably, A-DPS always initiates sampling from the center, whereas PGA-DPS begins with a mixture of central and peripheral lines, resulting in a more balanced coverage of frequency components.
>
> Furthermore, PGA-DPS significantly reduces the computational cost inherent to active sampling. Unlike A-DPS, which trains one network per target sample, PGA-DPS delivers comparable reconstruction quality with far fewer networks.
>
> >**W1. The key details are hidden away in the Appendix.**
>
> If there are any important details currently only in the Appendix, we would be happy to incorporate them into the main manuscript at the reviewer’s recommendation.
>
> >**typos**
>
> Thank you for pointing out the typos. We have corrected all typos.

---

> > ### Comment · Reviewer_yF5c · 2025-11-27
> >
> > Thank you for addressing the feedback. I have raised my scores accordingly.

---

> > > ### Author Response · Authors · 2025-11-28
> > >
> > > We sincerely appreciate your recognition of our work’s core contributions and strengths.
> > >
> > > If you have any further questions or suggestions, please feel free to reach out. We remain committed to providing any needed clarification.
> > >
> > > Thank you again for your constructive and thoughtful review.
> > >
> > >
> > > Best regards,
> > >
> > > The Authors

---

### Official Review · Reviewer_4uGe · 2025-11-01

**Soundness:** 3
**Presentation:** 3
**Contribution:** 2
**Rating:** 6
**Confidence:** 3

**Summary:**

The paper proposes PGA-DPS, a method that integrates a fixed prior sampling mask with group-based active sampling. The proposed approach builds upon DPS and A-DPS. It is evaluated across three tasks: MNIST, MRI reconstruction, and hyperspectral image segmentation.

**Strengths:**

- Extensive experiments are conducted across three tasks.

- The paper is well-written and easy to follow.

- The proposed method is supported by theoretical analysis.

- The approach is an effective variant of the existing DPS-based methods.

**Weaknesses:**

- For all experiments, the number of training epochs is fixed according to the implementation details. It is unclear how the performance would change if the best-performing checkpoint were selected instead of the final epoch model.

- In the MRI reconstruction experiments, one setup crops the slices to 208x208, while another uses 320x320 images. The reason for this inconsistency is not explained.

- The ratios for the fixed prior (Ps) and active sampling (As) are central to the proposed method but are heuristically chosen for each task. While the authors present results across multiple Ps and As settings for the MRI reconstruction task, similar analyses are missing for the classification and segmentation tasks. The segmentation task uses (80%, 20%), which differs from MRI, and the MNIST configuration also varies. The search space for these ratios is large, but no principled optimization strategy is provided.

**Questions:**

- How would the performance change if model selection were based on validation performance (best checkpoint) rather than the final training epoch?

- What is the rationale behind using different image resolutions (208x208 vs. 320x320) across MRI experiments?

- How sensitive is the proposed approach to Ps or As settings in MINST and segmentation tasks?

- Since the choice of Ps and As ratios is crucial, what is the justification for determining these values?

---

> ### Author Response · Authors · 2025-11-21
> **Response to Reviewer 4uGe (1/3)**
>
> We sincerely thank the reviewer for their constructive feedback and for recognizing the strengths of our work. We hope to address your comments effectively and further improve the quality of this paper.
>
> >**Q1. How would the performance change if model selection were based on validation performance (best checkpoint) rather than the final training epoch?**
>
> We evaluated the model selected based on validation performance (best checkpoint) and found that its performance is nearly identical to that of the model at the final training epoch. For example, in MNIST classification using PGA-DPS (60\%,20\%), the test accuracies of final epoch model and the best checkpoint are shown in Table R1.1, demonstrating the stability of our training framework.
>
> Table R1.1. The classification accuracy on the MNIST test set for various subsampling ratios.
> |                   | r=1   | 2     | 3     | 4     | 5     | 6     | 7     | 8     |
> |-------------------|-------|-------|-------|-------|-------|-------|-------|-------|
> | Final epoch model | 66.29 | 86.95 | 93.91 | 95.44 | 96.79 | 97.38 | 97.39 | 97.94 |
> | Best checkpoint   | 65.38 | 86.95 | 93.91 | 95.51 | 96.79 | 97.38 | 97.39 | 97.94 |
>
> In our experiments, the validation loss did not diverge for any of the tasks. With larger models or more training epochs, selecting the best checkpoint could provide greater benefit by preventing overfitting.
>
>
> >**Q2. What is the rationale behind using different image resolutions (208x208 vs. 320x320) across MRI experiments?**
>
> As the reviewer commented, we tested two scenarios for the MRI reconstruction task. First, to reduce the computational overhead associated with running extensive experiments across multiple Ps_ and _As_ configurations, we simplified the MR reconstruction task by using smaller images (208 × 208). Since our work focuses on proposing a novel sampling strategy that improves the efficiency of the sampling process itself rather than advancing the neural network architecture used for the downstream reconstruction task, we believe this simplification is an appropriate experimental choice.
>
> To validate this experimental setting and clinical applicability in more practical scenarios, we additionally conducted an MRI reconstruction experiment using the original k-space resolution (368X640) applying the best _Ps_ and _As_ configurations identified in 208x208 case. The evaluation metrics were computed using only the central 320×320 region to focus on the anatomically relevant structures.

---

> ### Author Response · Authors · 2025-11-21
> **Response to Reviewer 4uGe (2/3)**
>
> >**Q3. How sensitive is the proposed approach to _Ps_ or _As_ settings in MINST and segmentation tasks?**
>
> In the MNIST task, we evaluated multiple _Ps_ values under an _As_ of 20\% and multiple _As_ values under a _Ps_ of 60\%. Table R1.2 shows that PA-DPS outperforms the baselines in almost all configurations, although the optimal configuration shifts slightly across sampling ratios.
>
> In the HSI segmentation task, where the number of available spectral bands is extremely limited (5 bands, r ≈ 9.8\%) (Table R1.3.), the choice of _Ps_ and _As_ configurations is constrained. Thus, _As_ is fixed at 20\%. A small _Ps_ value in PGA-DPS makes its behavior similar to A-DPS, whereas a large Ps makes it closer to DPS. Consequently, under such extreme subsampling, the HSI task is sensitive to the behaviors of both A-DPS and DPS. Nevertheless, PGA-DPS consistently avoids inferior performance and remains competitive with, or better than, both baselines.
>
> We would like to emphasize that none of the previous studies have incorporated deep probabilistic sampling into hyperspectral imaging, which further highlights the contribution of our work.
>
> **Table R1.2.** The classification accuracy on the MNIST test set for various subsampling ratios (r) using various configurations of PGA-DPS (_Ps_, _As_). Each sampling strategy was repeated across six independent runs.
> | Sampling ratio (r)  =      | 1    | 2    | 3    | 4    | 5    | 6    | 7    | 8    | 100 |
> |---------------------|------|------|------|------|------|------|------|------|------|
> | DPS                 | 63.5 | 83.6 | 91.5 | 95.2 | 96.4 | 97.1 | 97.3 | 97.6 | 98.2 |
> | A-DPS               | 64.6 | 85.2 | 92.2 | 95.3 | 96.1 | 96.6 | 97.1 | 97.2 | -    |
> | PGA-DPS (80,20)     | 69.1 | 86.0 | 93.9 | 95.8 | 96.8 | 97.3 | 97.6 | 97.9 | -    |
> | PGA-DPS (60,20)     | 68.8 | 86.9 | 93.9 | 95.8 | 96.7 | 97.2 | 97.5 | 97.7 | -    |
> | PGA-DPS (40,20)     | 67.8 | 86.8 | 93.6 | 95.9 | 96.7 | 97.3 | 97.4 | 97.6 | -    |
> | PGA-DPS (20,20)     | 66.8 | 85.9 | 93.2 | 95.6 | 96.4 | 97.0 | 97.2 | 97.5 | -    |
> | PGA-DPS (60,5)      | 68.8 | 87.1 | 93.7 | 95.8 | 96.5 | 97.0 | 97.2 | 97.6 | -    |
> | PGA-DPS (60,10)     | 68.8 | 87.1 | 93.8 | 95.7 | 96.7 | 97.2 | 97.4 | 97.7 | -    |
> | PGA-DPS (60,15)     | 68.8 | 87.6 | 94.2 | 95.8 | 96.6 | 97.2 | 97.5 | 97.7 | -    |
> | PGA-DPS (60,25)     | 68.8 | 87.0 | 93.8 | 95.8 | 96.7 | 97.2 | 97.5 | 97.7 | -    |
> | PGA-DPS (60,30)     | 68.3 | 87.0 | 93.9 | 95.7 | 96.7 | 97.2 | 97.5 | 97.7 | -    |
>
> **Table R1.3.** Average segmentation results over 10 runs on the hold-out AeroRIT test set (3,127 patch samples) for band selection (using 5 bands, i.e. r ≈ 9.8\%), evaluated across various configuration settings of PGA-DPS (_Ps, As_).
> | Metric | A-DPS (0,20) | PGA-DPS (40,20) | PGA-DPS (60,20) | PGA-DPS (80,20) | DPS (100,0) |
> |--------|--------------|-----------------|-----------------|-----------------|-------------|
> | MPCA   | 65.55        | 81.29           | 82.59           | 86.37           | 85.75       |
> | mIOU   | 0.5181       | 0.6258          | 0.6554          | 0.6752          | 0.6703      |
> | mDICE  | 0.6158       | 0.7305          | 0.7572          | 0.7803          | 0.7732      |

---

> ### Author Response · Authors · 2025-11-21
> **Response to Reviewer 4uGe (3/3)**
>
> >**Q4. Since the choice of _Ps_ and _As_ ratios is crucial, what is the justification for determining these values?**
>
> This is an important question. As the reviewer noted, while PGA-DPS excels at generating subsampling strategies across diverse tasks, one potential limitation lies in selecting the hyperparameters that determine the proportion of fixed and active subsampling, denoted as _Ps_ and _As_, respectively.
>
> In our method, the optimal sampling portion varies slightly depending on the downstream task, task model, and the sampling ratio. We observed that an active sampling ratio (_As_) range of 20-30\% provides the best trade-off between the benefit of top-k sampling and the total budget allocated to active sampling. This observation substantially narrows the search space for selecting optimal _Ps_ and _As_ combinations. Additional experiments on the CIFAR-10 dataset also confirmed that _As_ within this range works well, as shown in Table 2 of the revised manuscript (See Section 4.2. CIFAR-10 Classification).
>
> To determine an appropriate prior-sampling ratio (Ps), we recommend using the performance discrepancy between DPS and A-DPS as an empirical indicator of the underlying Lipschitz characteristics of the task. Specifically, if DPS consistently outperforms A-DPS under a given configuration, this suggests that the objective landscape exhibits a larger effective Lipschitz constant—implying that a higher Ps would be beneficial. Conversely, if A-DPS achieves superior performance, the landscape is likely smoother (i.e., associated with a smaller Lipschitz constant), in which case a smaller Ps is preferable. For example, in the MRI reconstruction task, A-DPS outperforms DPS, and thus we used a small _Ps_ of 30 \%. In contrast, in the HSI segmentation task, DPS outperforms A-DPS, so we opted for a large _Ps_ of 80%.
>
> While we suggest a strategy for choosing optimal hyperparameter of (_Ps, As_), automatically tuning them (Bergstra et al., 2011; Feurer & Hutter, 2019; Bischl et al., 2023) could further improve the robustness of the proposed method. Future work may explore joint optimization of the active sampling hyperparameters (_Ps, As_), the subsampling trajectory, and the downstream task model. We have added this discussion to the “LIMITATION & FUTURE WORK” section in the revised version of the main paper.

---

### Meta-Review · Area_Chair_HCPw · 2026-01-07

**Summary:**

- This paper proposes a novel method for subsampling, which is evaluated across three different vision tasks.
- The reviewers all find the paper well-written and appreciate the significant empirical improvement provided by the proposed method. Also, the reviewers all praise the various tasks and baselines considered in this paper.

- The reviewers also acknowledge the theoretical guarantees provided in this paper.

- Some reviewers question the sensitivity of the proposed method. The rebuttal provides additional numerical results to demonstrate the low sensitivity.

- Three reviews were collected with scores of 6,4, 4, but both reviewers who gave scores 4 have decided to raise their scores during the discussion.

**Reviewer Concerns:**

Solved concerns

- The reviewers ask some clarifying questions on the experiment settings, which are successfully addressed by the rebuttal.

- Reviewer 4uGe asks about how the performance change if the model selection were based on validation performance. The rebuttal provides additional simulation results to show that the outcome is quite similar.

- All reviewers ask about the sensitivity of the proposed method. The rebuttal provides additional simulation results to show the low sensitivity.

- Reviewer yF5c's major concerns are the novelty of the methodology compared with [1] and whether Theorem 1 is a known result. The rebuttal clearly explains the differences between the proposed method and the existing one conceptually, and clarifies that theorem 1 is a new result.

- Reviewer 9EHq also questions the limited dataset scale used in the paper. The rebuttal provides additional simulation results on larger datasets.

During the discussion, the reviewers find the authors addressed their concerns successfully and have raised the score accordingly.

**Reviewer Scores:**

Three reviews were collected with scores of 6,4, 4. The reviewer who gave 6 is not likely to change the score. The two reviewers who gave scores 4 have decided to raise their scores during the discussion.

---

### Decision · Program_Chairs · 2026-01-26

Accept (Poster)